# The Effect of Confined Granular Soil on Embedded *PZT* Patches Using *FFT* and Digital Static Cone Penetrometer (DSCP)

**Nisha Kumari** *[ID] **and Ashutosh Trivedi** [ID]

Civil Engineering Department, Delhi Technological University, Delhi 11042, India
* Correspondence: nishasoni.ce@gmail.com

**Abstract:** This paper presents the effect of confined granular fill on the efficiency of embedded *PZT* (Lead Zirconate Titanate) patches. The experiments are carried out to capture the dynamic response of confined granular fill and voltage output. The dynamic response of the system is analysed using the fast Fourier transformation (*FFT*) of vibration and digital static cone penetrometer (DSCP) test at various excitation frequencies. The peak pick method is used to determine the damping loss factor and natural frequency. The efficiency of voltage generation of embedded *PZT* patches is influenced by the depth of embedment, excitation frequency, location, and engineering properties of the confined granular fill. The results show that the influence of vibration frequency and placement of *PZT* on the output voltage should be considered. The thickness and engineering properties of granular soil affect the efficiency of voltage output. This study emphasizes the utility of *PZT* patches in the subgrade and structural health monitoring of various geotechnical structures.

**Keywords:** granular soil; *PZT* patch; excitation frequency; *FFT*; digital static cone penetrometer; voltage output





## 1. Introduction

Advances in smart infrastructure and the rise in demand for energy have led to generating energy sources utilizing mechanical vibrations. Piezoelectric energy harvesting has the potential to convert mechanical vibration into electrical energy. Substantial research has been embarked upon to elucidate the influence of the configuration of a piezoelectric energy harvester (*PEH*) on its efficiency of energy conversion [1–5]. Kim et al. [2] analysed the Cymbal *PEH*, which can generate power (1.2 mW) at 20 Hz for each passing vehicle. Wang et al. [4] investigated stack piezoelectric devices installed on a railway track. The results show that the energy increased to 1000 mJ when the stiffness increased to a significantly high value (MN/m). Subsequently, the electrical voltage was generated at 97 V (0.06 J) per vehicular movement on asphalt pavement. Zhang et al. [5] investigated the performance of a cantilever *PEH* for harvesting energy from bridge vibrations. The *PEH* was positioned where the mode shape had a peak displacement. As the fundamental frequency of the *PEH* was closest to one of the bridge's natural frequencies, more energy could be harvested. The previous studies show that the energy harvester's configuration and vibration intensity affect the output of *PZT* patches.

The principle of energy conversion using piezoelectric materials depends on the dynamic response of the input vibration [6,7]. Peigney and Siegert [8] concluded that the amplitude and frequency of vibrations of a bridge influence the power output. A few researchers [9,10] investigated the effectiveness of embedded piezoelectric elements. They developed a compression model to analyse the efficiency of power output. They indicated that the output power of piezoelectric material depends on the loading cycles on the asphalt pavements. The analysis of excitation force, damping ratio, and natural frequency of the structure need to be considered to enhance energy conversion. However, none of these previous studies considered the effect of abutting material on the efficiency of the *PZT* patches when embedded in the pavement.

Despite abutting material properties, the output efficiency of the *PZT* patch is also influenced by the frequency of mechanical vibrations. Roshani et al. [11] indicated that the amplitude and frequency of the loading condition significantly affect the output power. Cahill et al. [12] investigated the energy harvesting from the bridge under varied loading conditions. The energy harvester showed a peak voltage response at the natural frequency. Ding et al. [13] studied the effect of piezoelectric energy harvesting under traffic load vibrations. They investigated the influence of vibration frequency and excitation load on power output.

Kumari and Trivedi [14] studied the influence of confined granular fill on horizontally and vertically embedded granular fill. They concluded that the alignment of the *PZT* patch influences the voltage output. The research indicates that the dynamic response of the system significantly affects energy harvesting. These studies considered the effect of the pavement surface and traffic loads on the efficiency of *PEHs*. They did not consider the influence of subgrade on the performance of the piezoelectric energy harvesters. However, the influence of various subgrade materials and engineering properties on converting vibration energy for *PZT* patches under various excitation frequencies has not been clearly understood.

This paper aims to explore an experimental analysis of the influence of the abutting material's properties and the excitation frequency on the voltage output of *PZT* patches. Some experiments were conducted on confined granular fill at varying depths to analyse the effect of embedment depths. This paper shows the effect of both vertical and transverse mechanical vibrations of varied excitation frequencies. The dynamic responses of the experimental data were analysed using fast Fourier transformation (*FFT*) in the frequency domain. The system's natural frequency was investigated experimentally using the peak pick method from the frequency response (*FRF*) function of the system. An analysis of the influence of excitation frequency, embedment depth, and displacement on voltage output is presented based on experimental data. Further, a set of static cone penetration tests was carried out to monitor the effect of confinement under various excitation frequencies. This effort emphasizes the utility of *PZT* patches in granular fills with vibration and their uses in piezoelectric energy harvesting from the ambient vibration from civil structures.

## 2. Material and Methods

The granular material used in the present work is classified as poorly graded sand using the particle-size analysis as per the Indian Standard Soil Classification system (IS: 2720). The grain size distribution curve of the granular soil is shown in Figure 1. The index properties, as determined in the laboratory, are given in Table 1.

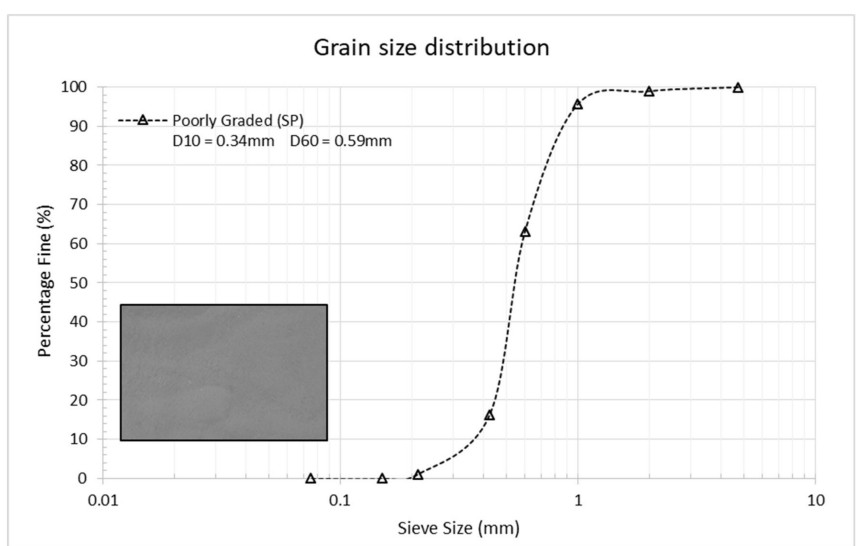

**Figure 1.** Particle size distribution of granular material (grey in colour) used in the present study.

**Table 1.** Index properties of the confined granular materials.

| Maximum Dry Unit Weight (kNm$^{-3}$) | Degree of Compaction (%) | Specific Gravity | Optimum Moisture Content (%) | IS Classification |
|---|---|---|---|---|
| 18.4 | 96.7 | 2.5 | 13.91 | SP |

*Piezoelectric Materials*

Piezoceramic (*PZT*) materials have been widely applied to convert mechanical energy into electrical energy in engineering applications [6,8,15,16]. The constitutive equations of *PZT* materials as per IEEE (1987) are expressed as

$$S_P = S_{pq}^E T_q + d_{kp} E_k \tag{1}$$

$$D_i = d_{iq} T_q + \xi_{ik}^T E_k \tag{2}$$

where $S_p$ represents the mechanical strain tensor; $D_i$ denotes the electric displacement; $T_q$ is stress tensor; $d_{kp}$ is strain coefficient matrix; $E_k$ is applied external electric field vector; $S_{pq}^E$ is elastic compliance coefficient at the constant electrical field; and $\xi_{ik}^T$ dielectric permittivity at constant stress.

The *PZT*-soil coupled system can be modelled as a 1-D system as the constraint along the thickness direction of the *PZT* patch. In the vertical direction, assuming the polarization is along the *z*-axis. In the transverse direction, the polarization is along the *x*-axis. The *PZT* material is considered an isotropic material in the *x*-*y* plane. Thus, the constitutive equations (Equations (1) and (2)) of the *PZT* patch are expressed as,

$$S_{ZP} = S_{33p}^E T_{ZP} + d_{33} E_Z; \; S_{xP} = S_{31p}^E T_{xP} + d_{31} E_x \tag{3}$$

$$D_Z = d_{33} T_{ZP} + \xi_{33}^T E_Z; \; D_x = d_{31} T_{xP} + \xi_{31}^T E_x \tag{4}$$

The stress–strain in soil material along the *z* and *x*-direction can be expressed as follows:

$$\varepsilon_{ZS} = S_{33S}^E T_{ZS}; \; \varepsilon_{xS} = S_{31S}^E T_{xS} \tag{5}$$

where $\varepsilon_{ZS}$ and $T_{ZS}$ are the stress and strain vectors of the granular soil along the *z*-direction, respectively. $\varepsilon_{xS}$ and $T_{xS}$ are the stress and strain vectors of the granular soil along the *x*-direction, respectively. $S_{33S}^E$ represents the elastic compliance coefficient of the granular soil.

The geometric relationship equations of the *PZT* patch and granular soil composite system [17] are expressed as,

$$\varepsilon_{ZS} = \varepsilon_{PS} = \frac{\partial w_{PS}}{\partial Z}; \; \varepsilon_{xS} = \varepsilon_{PS} = \frac{\partial w_{PS}}{\partial X} \tag{6}$$

$$E_Z = -\frac{\partial \phi}{\partial Z} = 0; \; E_x = -\frac{\partial \phi}{\partial X} = 0 \tag{7}$$

where $w_{PS}$ is the displacement of the *PZT* patch and soil composite in the *z* direction, and $\phi$ is the electric potential. In the present study, the external electric field is considered negligible.

## 3. Experimental Set-Up and Procedure

### 3.1. Test Model

To study the effect of various parameters, such as the exciting frequency and depth of the *PZT* patch, required conducting an experiment. Therefore, a laboratory set-up was designed to investigate the effect of the natural frequency of the confined granular fill system on *PZT* patches in a steel tank of 600 × 460 mm with 400 mm depth. The experimental set-up included the steel tank, granular material, dynamic shaker, *PZT* patches, piezoelectric accelerometers (sensors), *FFT* analyser, and oscilloscope (Figure 2). The vibration tests were performed through a steel plate of size (270 × 230 mm) and

thickness (10 mm). The tank was filled with compacted granular soil in three layers to maintain the uniform density of 1785 kg·m$^{-3}$ throughout the depth. It was assumed that the dynamic response of the soil is independent of the plate thickness. A *PZT* patch of diameter 27 mm was used. In solving the problem related to dynamic response analysis in the frequency response domain, fast Fourier transform is an efficient method for both single- and multi-degree freedom systems [18]. Therefore, all data were collected as the *FFT* of the vibrations at various excitation frequencies. The layout of the embedded *PZT* patches in confined granular fill is shown in Figure 3.

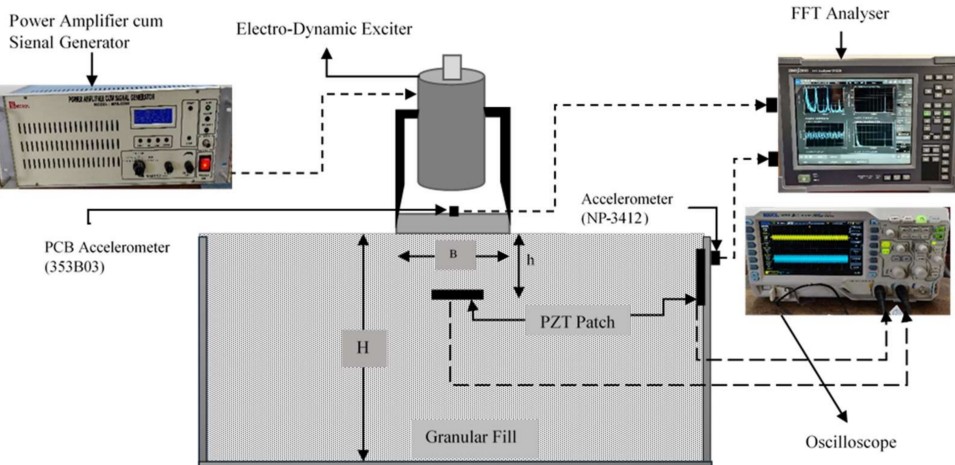

**Figure 2.** Laboratory experimental set-up for dynamic response and voltage measurement.

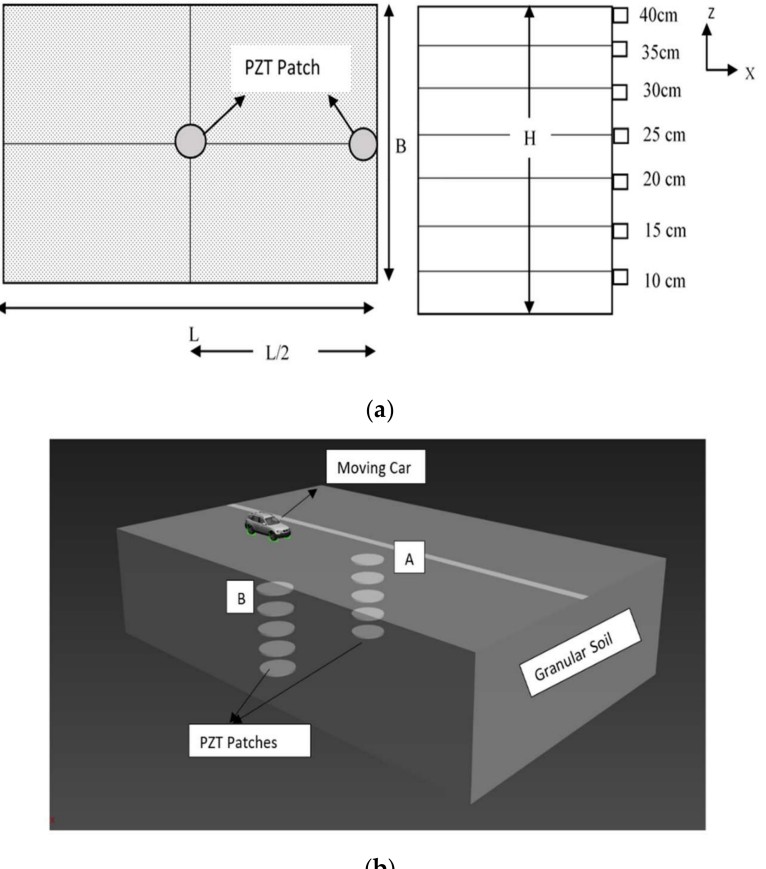

**Figure 3.** (**a**) Layout of the PZT patch embedded into the confined granular fill for the vertical and transverse vibrations; (**b**) The optical image of the embedded PZT patches in pavement at various depth.

### 3.2. Electro-Dynamic Vibrator and Instruments

An electro-dynamic exciter (Uttrakhand, India) (MEV-0020) was used to produce vibration. The system generated the vibrations with the drive coil connected rigidly to the moving platform and positioned. When AC flows in the drive coil, it gives rise to force by converting an electric current into mechanical vibrations that move the platform. The exciter was connected to the power amplifier cum signal generator (MPA-0500) to control the mechanical force. The function of a power amplifier cum signal generator was used to generate and amplify the signal sufficiently to drive the exciter to the desired frequency.

The dynamic response of the confined granular soils was monitored using two accelerometers, one PCB piezoelectric accelerometer (353B03) with low noise cable (M003), and one accelerometer (NP-3412) with signal cable (S381) (Haryana, India) (Figure 2). The PCB (353B03) has a sensitivity of 9.98 mV/g, transverse sensitivity of 0.9%, and resonant frequency of 56.1 kHz. The accelerometer NP-3331 has a sensitivity of 5.038 mV/g. Both sensors were connected to an *FFT* analyser (CF-9200). The *FFT* analyser has a bandwidth of 100 mHz to 100 kHz and a frequency tolerance of ±0.005%. The PCB accelerometers were attached to the centre of the granular fill to measure the vertical vibration. An accelerometer (NP-3412) was employed on one side of the tank to monitor the horizontal vibration. The data collected were the *FFT* spectrum and *FRF* (frequency response function) of vibration for various exciting frequencies.

To measure the effect of vibration on the voltage output of piezoelectric patches, the positive and negative node of the *PZT* is connected to an oscilloscope (DS1102Z-E) using the probes (PVP 3150). The output data are obtained in the form of voltage vs. time.

### 3.3. Modal Parameter Estimation

The peak picking modal estimation technique was utilized in natural frequency analysis. It has been commonly used due to its simplicity. The dynamic response of a system to an external excitation load is described as

$$M\ddot{x}(t) + C\dot{x}(t) + Kx(t) = F(t) \tag{8}$$

where $M$, $C$, and $K$ are matrices of mass, damping, and stiffness of the system, and $x$, $\dot{x}$, and $\ddot{x}$ represent the displacement, velocity, and acceleration response of the system, respectively.

The peak picking method is considered for estimating the modal features of a dynamic system based on response data. This approach could characterize ambient vibration excitation as standard white-noise Gaussian distribution based on the fact that the frequency response function (FRF) of a system would experience peak values in the frequency domain. In the case of a white noise excitation, the *FRF* of a system at sensor location k, $H_k = (jw)$, is considered equivalent to the Fourier spectrum of the response data gathered by the sensor. This range is derived by converting the measured response to the frequency domain using *FFT*. Therefore, the natural frequency of the system is the dominant frequency of the *FRF* of the system [19].

The damping loss factor is analysed using the frequency response function expressed as follows,

$$\left|\frac{x}{F}\right| = \frac{1/k}{\sqrt{1 - \left(\frac{\omega^2}{\omega_n^2}\right)^2 + \frac{4\zeta^2\omega^2}{\omega_n^2}}} \tag{9}$$

where $\zeta$ is the damping loss factor, the natural frequency of the system $\omega_n = \sqrt{k/m}$, and the spring constant $k$. $\omega$ is excitation frequency.

The measurements of modes at resonances are difficult for a lightly damped system. Simultaneously, the response of a heavily damped system is significantly influenced by more than one mode. The mode shapes are determined by the peak picking method using the system's FRFs.

### 3.4. Digital Static Cone Penetrometer

A digital static cone penetrometer (SCP) test is a low-cost method for measuring the penetration resistance of fine granular materials during site investigation and laboratory research. The cone tip was hydraulically forced into the soil during the SCP test at a consistent penetration rate. The cone tip resistance and sleeve friction resistance were measured. In this study, a digital SCP was used, which consists of a measuring body with two push handles, a drive rod with a diameter of 16 mm and a length of 498 mm, a 60° cone (area of cone base $A_c = 9.97$ cm$^2$) assembled at the bottom of the drive rod, a linear variable displacement transducer (LVDT) with a range of 0–200 mm, a data acquisition system with a USB output slot, and a load cell with a capacity of 300 kg, as shown in Figure 4. It also depicts the layout of the test sections in the soil tank.

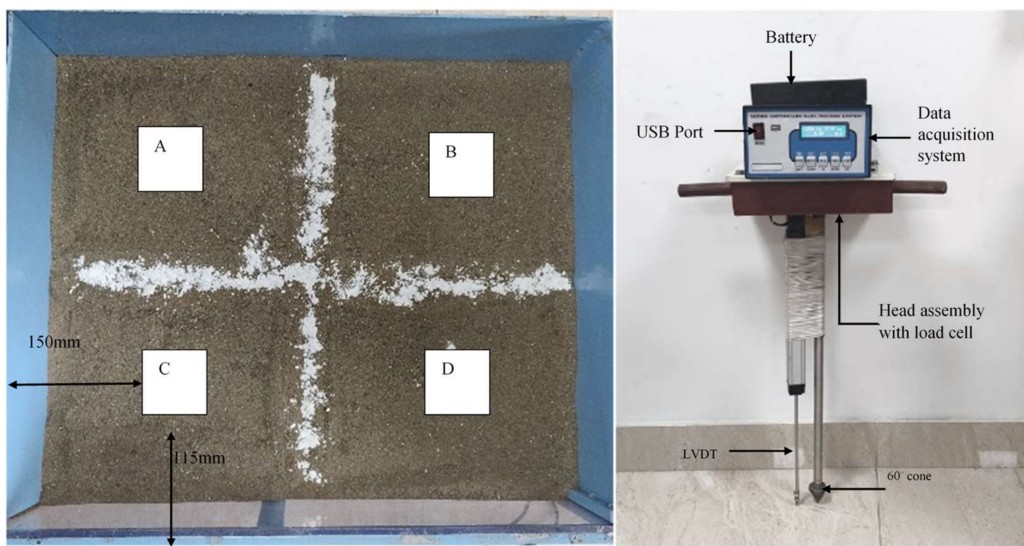

**Figure 4.** Layout of the test section measurement and testing locations A, B, C and D of digital SCP test.

The digital SCP is handheld and capable of penetrating sub-base and subgrade soils. The push handles are provided at the head assembly of the digital SCP. It penetrates the cone attached to the driving rod into the soil at a constant force given with the help of body weight. During the testing, both the load cell and the LVDT were active. The load cell assists in measuring the applied load, while the LVDT measures the digital SCP penetration. Both provide the electrical signal to the data acquisition system, transforming it from analogue to digital. The digital SCP was positioned vertically on the surface of the granular soil. Jerks were avoided while pushing the handles of the digital SCP since they affect the output data. The data from the load cell and LVDT were automatically saved in a tabular format with two columns, load and displacement, in the USB output device. When the applied force causes a negligible change in the magnitude of penetration, the device records refusal. It may produce inconsistent readings and damage the cone, driving rod, and load cell.

## 4. Results and Discussion

The experimental study investigated the dynamic responses of the confined granular fill with different excitation frequencies. The vibration generated due to harmonic excitation in a frequency range 10–50 Hz was collected in the form of a power spectrum and frequency response function. The properties of materials used as *PZT* patches are listed in Table 2.

**Table 2.** Properties of the *PZT* patch.

| Properties | Symbol | Magnitude |
|---|---|---|
| Dielectric constant | $\frac{\varepsilon_{33}^T}{\varepsilon_o}$ | 3270 |
| Dielectric loss | $\tan\delta$ | 0.016 |
| Electromechanical coupling factor | $k_{31}$ | 0.52 |
| Piezoelectric charge constant (pC N$^{-1}$) | $d_{31}$ | −275 |
| Piezoelectric voltage constant ($10^{-3}$ Vm N$^{-1}$) | $g_{31}$ | −9.5 |
| Young's modulus (GPa) | Y | 61 |
| Unit weight (kNm$^{-3}$) | $\gamma$ | 7.2 |

### 4.1. Natural Frequency Analysis

Fast Fourier transformations (FTT) were utilized to obtain the damping characteristics and natural frequency of the system in the frequency domain. The piezoelectric accelerometers were attached at various depths ($H$ = 20 cm, 25 cm, 30 cm, 35 cm, and 40 cm) to analyse the dynamic responses of the confined granular fill. The arrangement of the accelerometer is shown in Figure 3. The confined granular fill was subjected to vibrations of various frequencies ($f$ = 10–50 Hz). The peak amplitudes of the acceleration, velocity, and displacement response at various excitation frequencies are shown in Table 3 for both vertical and transverse vibrations.

**Table 3.** Peak amplitude of vibration in vertical and transverse direction for confined granular fill.

| Input Excitation Frequency (Hz) | Vertical Vibration | | | Transverse Vibration | | |
|---|---|---|---|---|---|---|
| | Acceleration ($\mu$m/s$^2$) | Velocity ($\mu$m/s) | Displacement ($\mu$m) | Acceleration ($\mu$m/s$^2$) | Velocity ($\mu$m/s) | Displacement ($\mu$m) |
| 10 | 2579 | 52 | 0.537 | 1243 | 20 | 0.290 |
| 20 | 913.26 | 0.759 | 0.007 | 77.128 | 1 | 0.008 |
| 30 | 272,000 | 1441 | 5.90 | 815,000 | 4389 | 18.33 |
| 40 | 103,000 | 233 | 0.93 | 57,920 | 299 | 1.19 |
| 50 | 2102 | 46 | 0.023 | 1766 | 45 | 0.018 |

The peak response (acceleration, velocity, and displacement) of the confined granular fill at the same position varied directly as the excitation frequency. A similar observation was made by Ding et al. (2018) for the dynamic response of the subgrade at various exciting frequencies. The peak response of the confined granular at centre point is greater than the point to the right; due to the boundary effect. The geometric law was followed to understand the relationship between stress, strain, and displacement. The displacement spectrum for various excitation frequencies characterizes the vibrations generated at different depths.

Figure 5 shows the displacement response spectrum of the system at various excitation frequencies ($f$ = 10–50 Hz) at varying depths of the confined granular fill. This spectrum was obtained using the *FFT* analyser and accelerometers. The purpose of obtaining the system's response spectrum is to analyse the damping loss factor and natural frequency of the system. The displacement has a maximum at the natural frequency for the excitation frequencies of 10–30 Hz. In this case, the peak is at 31 Hz. For high frequencies, the peak displacement tends towards 40 Hz. At high frequencies, the confined system will behave as a rigid body, so the mass just follows the upper layer motion. Figure 5E depicts that the peak displacement of the vibration was obtained at the 40 cm thickness, whereas for low frequencies (10–30 Hz), the maximum displacement obtained was closer to 30 cm. The displacement response also represents the intensity of the vibration at a deeper depth to understand the effect of vibration on the *PZT* patches embedded at various thicknesses of the granular fill.

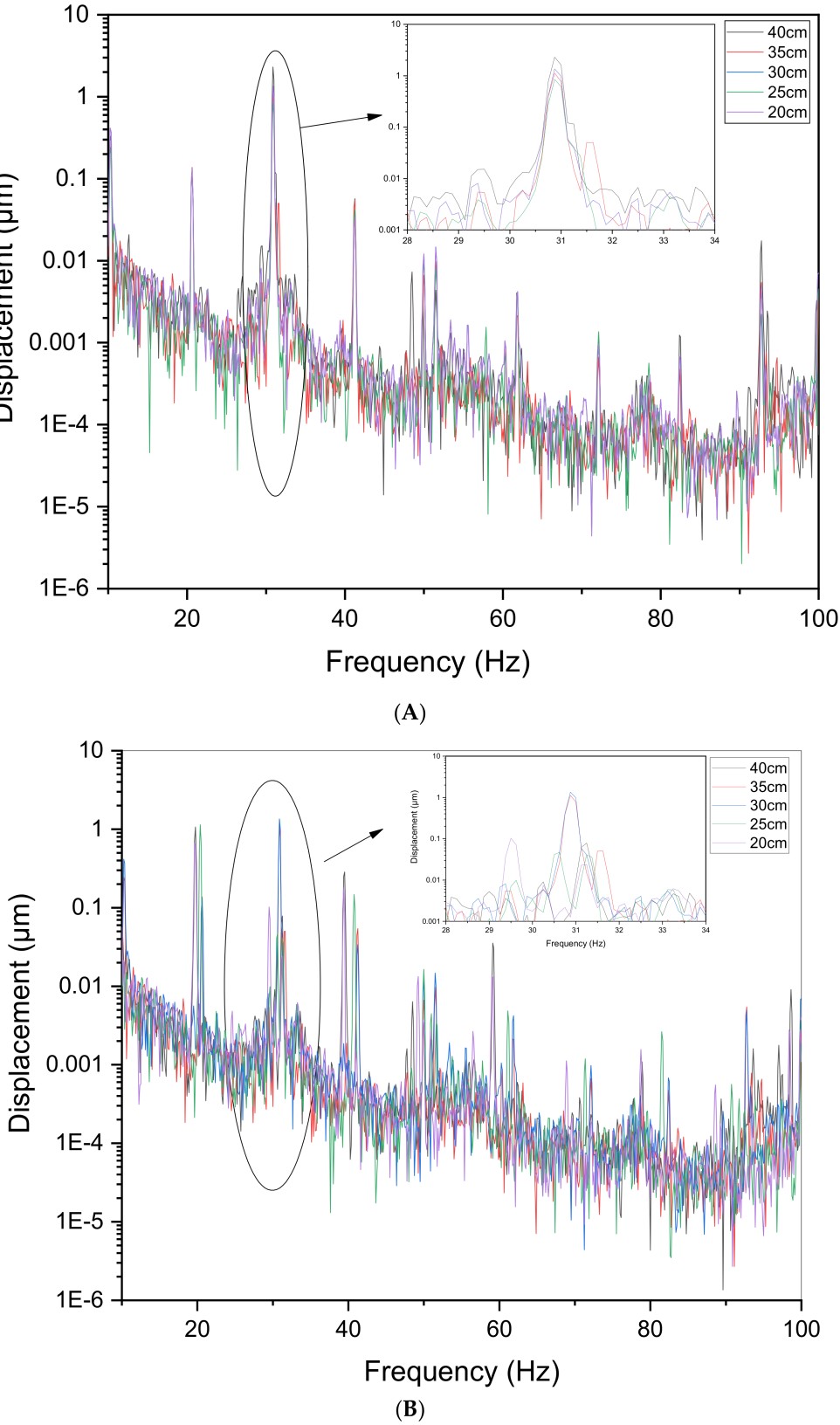

**Figure 5.** *Cont.*

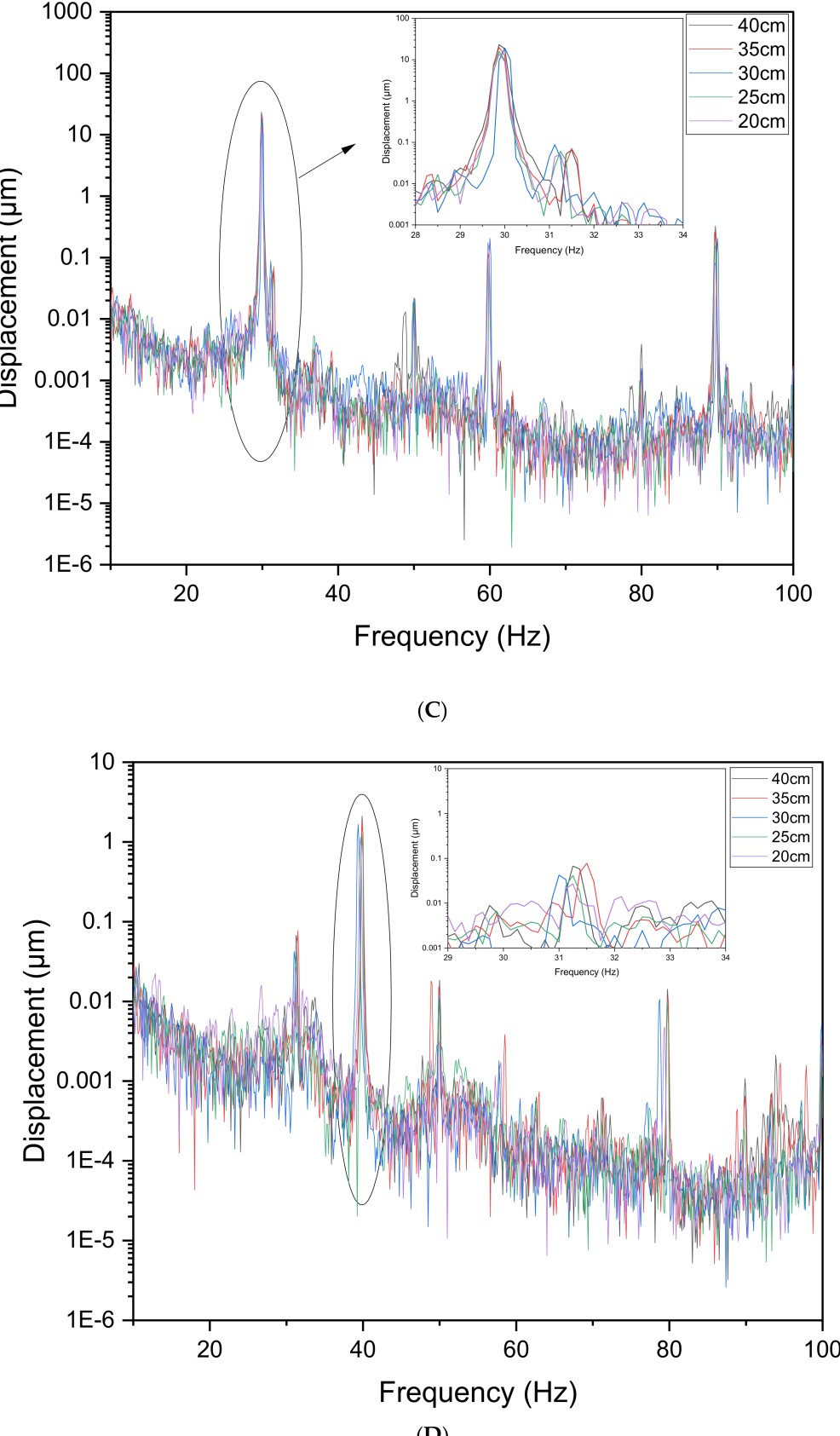

**Figure 5.** *Cont.*

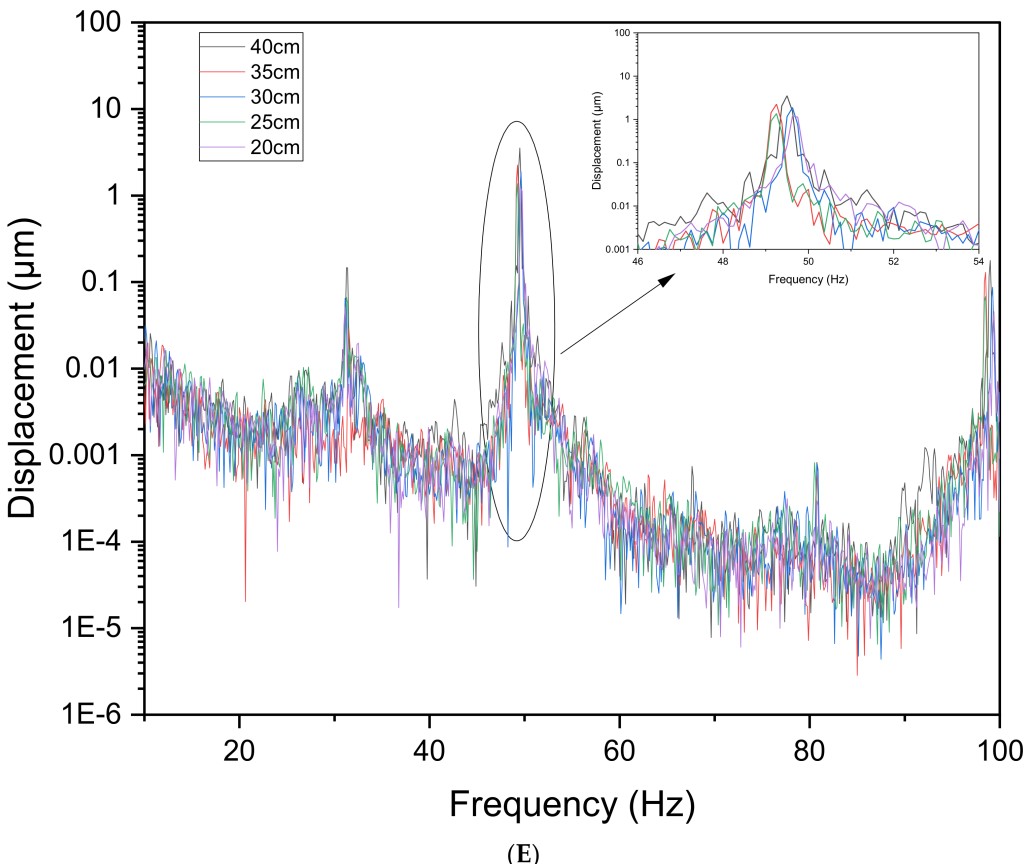

(**E**)

**Figure 5.** Influence of depth of granular fill (20–40 cm) on the displacement spectrum at an excitation frequency of (**A**) 10 Hz; (**B**) 20 Hz; (**C**) 30 Hz; (**D**) 40 Hz; (**E**) 50 Hz.

The natural frequency and damping loss factor were analysed using the peak pick modal parameter method from the frequency response function (FRF) of the experimental data. Table 4 represents the damping loss factors at various excitation frequencies.

**Table 4.** Damping loss factor and natural frequency of the confined granular fill.

| Excitation Frequency (Hz) | 10 | 20 | 30 | 40 | 50 |
|---|---|---|---|---|---|
| Natural frequency (Hz) | 31 | 31 | 30 | 32 | 49 |
| Damping loss factor ($\xi$) | 0.121 | 0.127 | 0.132 | 0.176 | 0.122 |

### 4.2. Cone Resistance of Confined Granular Fill

Figure 6A presents the cone resistance of filled granular soil subjected with and without external dynamic vibrations. Digital SCP tests were performed at locations 'A', 'B', 'C', and 'D' in the test section. Singh et al. [20] performed a digital SCP test to estimate the load displacement of fine-grained soils. Digital SCP results are not identical at all the testing locations. However, they follow a similar pattern for each test section at different excitation frequencies. The cone resistance can be utilized to determine the stress at a penetration depth. The results clearly show that after 10–20 Hz frequency vibration, granular soil has greater resistance against applied load than before vibration. After 30 Hz frequency vibration, the cone resistance decreased up to 22% due to changes in the relative density of the granular fill. The relative density and overburden pressure control the cone resistance of the granular fill [21]. Figure 6B presents the cone resistance of the granular fill after 40–50 Hz excitation frequency.

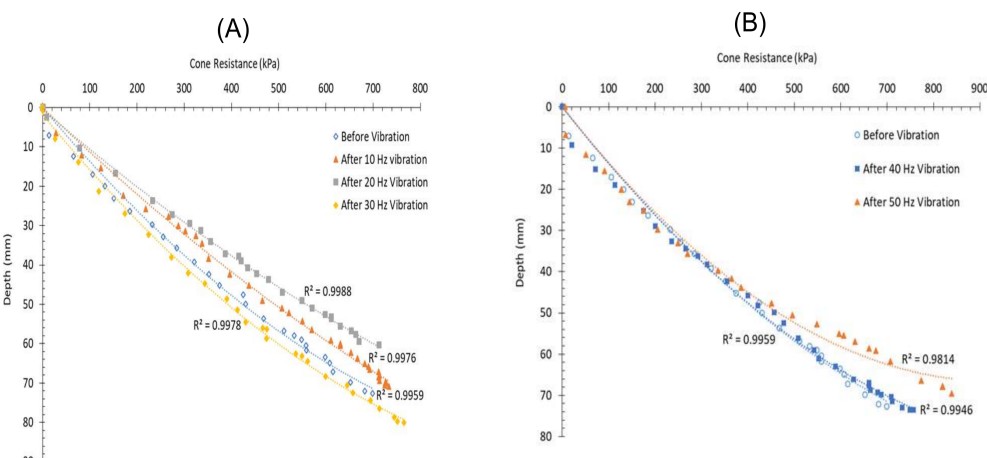

**Figure 6.** Cone resistance obtained from digital SCP data for confined granular fill with and without excitation frequency of (**A**) 10–30 Hz; (**B**) 40–50 Hz.

It is observed that the cone resistance increases after excitation frequency above the natural frequency. The overlapping at shallow depth signifies an unstructured response at low overburden pressure. The relationship between the cone resistance and penetration depth at varying excitation frequencies followed an empirical law as

$$D_P = \alpha q_c^2 + \beta q_c \tag{10}$$

where $\alpha$ and $\beta$ are fitting parameters that vary with an excitation frequency of vibration (Table 5) with a satisfactory coefficient of regression ($R^2$). The cone resistance $q_c$ and depth of penetration $D_p$ are expressed in kPa and mm, respectively.

**Table 5.** Fitting parameters $\alpha$ and $\beta$ for the confined granular fill at varying excitation frequency.

| Exciting Frequency (Hz) | $\alpha$ | $\beta$ | $R^2$ |
|---|---|---|---|
| 10 | $-3 \times 10^{-5}$ | 0.1157 | 0.9976 |
| 20 | $-3 \times 10^{-5}$ | 0.1035 | 0.9988 |
| 30 | $-7 \times 10^{-5}$ | 0.1539 | 0.9978 |
| 40 | $-6 \times 10^{-5}$ | 0.1439 | 0.9946 |
| 50 | $-8 \times 10^{-5}$ | 0.1432 | 0.9814 |

*4.3. Analysis of Output Voltage*

Figure 7 plots the variations in the voltage output of the *PZT* patch with respect to the excitation frequency at different embedment depths of *PZT* in the vertical direction ($f$ = 10–50 Hz, H = 40 cm, $h_r$ = 0.5–1). The variation in maximum output voltage at varied excitation frequencies is due to the displacement of granular soil. Compared with the other excitation frequency, it is found that maximum voltage output in the vertical direction was obtained for the depth ratio of 0.5 at resonance. Thus, a greater output voltage is obtained if the *PZT* patch is arranged at an appropriate depth from the surface of the granular soil. In other words, the *PZT* patch cannot be placed at a greater depth ratio and shallow depth in granular soils, as this reduces the voltage output efficiency. The efficiency of voltage output for the *PZT* patch embedded in a vertical direction at various excitations is shown in Table 6. The efficiency of the voltage output at varied excitation frequencies is analysed using the formula

$$\left[ \eta = \left( v_{f_1} - v_{f_2} \right) / \left( v_{f_1} \right) \right] \tag{11}$$

where $V_{f_1}$ is voltage output of *PZT* patch at 10 Hz frequency and $V_{f_2}$ is the voltage output of the *PZT* patch at various frequencies (20–50 Hz).

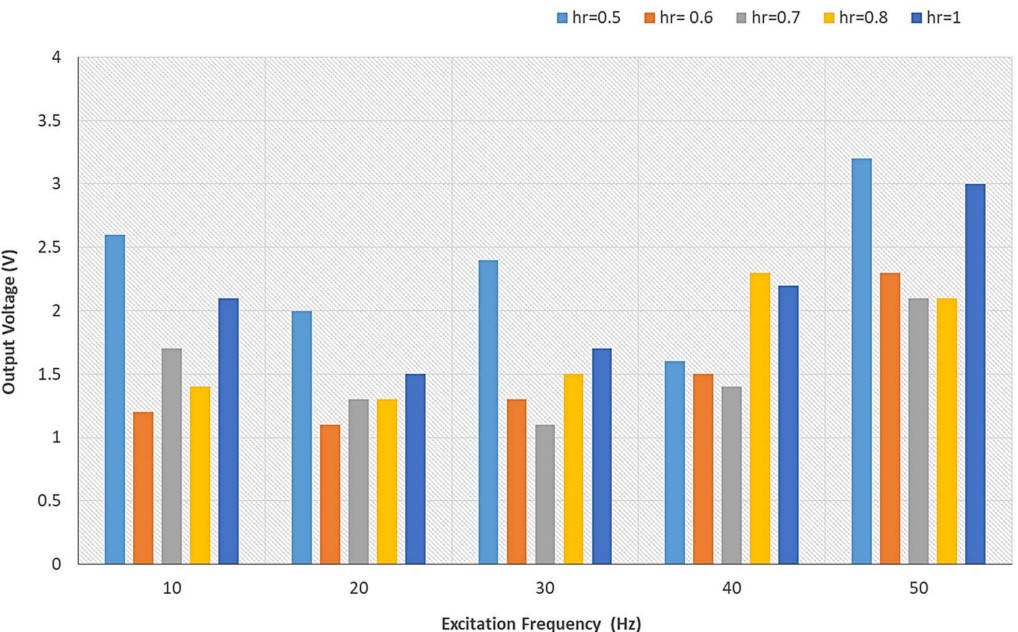

**Figure 7.** Output voltage for *PZT* patches embedded at varying depths in the vertical direction.

**Table 6.** Voltage output efficiency of *PZT* patch embedded in poorly graded soil for various excitation frequencies.

| Depth Ratio (hr) | Input Excitation Frequency (Hz) | Voltage Output in Vertical Direction ($V_v$) | Efficiency (%) | Voltage Output in the Transverse Direction ($V_t$) | Efficiency (%) |
|---|---|---|---|---|---|
| 0.5 | 10 | 2.6 | | 2.3 | |
| | 20 | 1.2 | 54 | 2.2 | 4.3 |
| | 30 | 1.7 | 35 | 2.4 | 4.3 |
| | 40 | 1.4 | 46 | 1.9 | 17.4 |
| | 50 | 2.1 | 19 | 2.2 | 4.3 |
| 0.6 | 10 | 2 | | 2.4 | |
| | 20 | 1.1 | 45 | 1 | 58 |
| | 30 | 1.3 | 35 | 2.6 | 8 |
| | 40 | 1.3 | 35 | 0.9 | 63 |
| | 50 | 1.5 | 25 | 1.7 | 29 |
| 0.7 | 10 | 2.4 | | 1.9 | |
| | 20 | 1.3 | 46 | 1.2 | 37 |
| | 30 | 1.1 | 54 | 1.5 | 21 |
| | 40 | 1.5 | 38 | 1.6 | 16 |
| | 50 | 1.7 | 29 | 1 | 47 |
| 0.8 | 10 | 1.6 | | 2.4 | |
| | 20 | 1.5 | 6 | 2.8 | 17 |
| | 30 | 1.4 | 13 | 2.2 | 8 |
| | 40 | 2.3 | 44 | 2.3 | 4 |
| | 50 | 2.2 | 47 | 2 | 17 |
| 1 | 10 | 3.2 | | 2.2 | |
| | 20 | 2.3 | 28 | 2.5 | 14 |
| | 30 | 2.1 | 34 | 2 | 9 |
| | 40 | 2.1 | 34 | 2.1 | 5 |
| | 50 | 3.0 | 6 | 1.9 | 14 |

Figure 8 shows the voltage output variation of the *PZT* patch with respect to the excitation frequency at different embedment depths in the transverse direction ($f$ = 10–50 Hz, $H$ = 40 cm, $h_r$ = 0.5–1). This attempt was made to analyse the effect of vibration on *PZT* patch embedded at a distance in lateral direction from the centreline. Table 6 shows the

voltage output efficiencies of *PZT* patches at various exciting frequencies embedded in a lateral direction. It can be observed from Figure 8 that the output voltage was minimum at 30 Hz excitation frequency and maximum at higher frequencies for a depth ratio of 0.6. Thus, the maximum voltage output is obtained at more than 30 Hz excitation frequency in the transverse direction. The variation in the output voltage at varied excitation frequencies is due to the confinement of the granular fill. Hence, the distance from the centreline of the load influences the efficiency of the *PZT* patch embedded in confined granular fill, similar result was found by Ding et al. (2018).

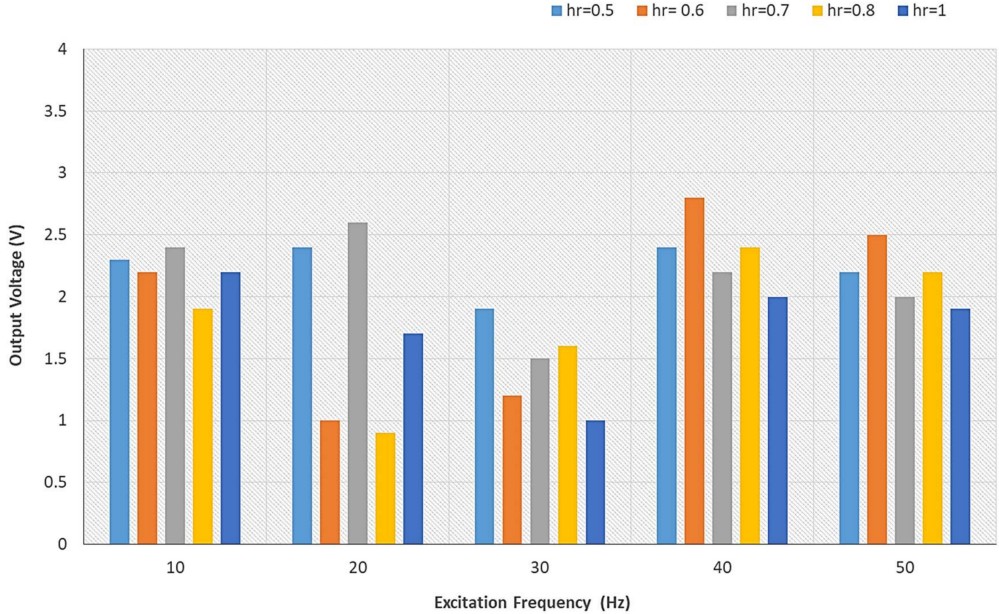

**Figure 8.** Output voltage for *PZT* patches embedded at varying depths in the transverse direction.

## 5. Conclusions

This study investigated the effect of the confined granular soil on the efficiency of the *PZT* patches under various exciting frequencies. The dynamic response of experimental data were analysed using fast Fourier transformation (*FFT*). The peak pick method was used to analyse the natural frequency of the system. An analysis was also performed of the influence of excitation frequency and embedment depth on the voltage output of *PZT* patches. Further, a set of static cone penetration tests were carried out to monitor the effect of confinement under various excitation frequencies. The significant findings are summarized as follows:

(i)   The voltage output from the *PZT* patch depends on the excitation frequency and placement of the *PZT* patch. For the same embedment depth, the peak voltage output in vertical direction is observed at 50 Hz, while in the transverse direction, it is obtained at 10 Hz.

(ii)  The depth of the compressive and tensile regions of the embedded *PZT* patch should be considered to obtain the maximum output voltage. For the same excitation frequency ($f$ = 10–20 Hz), the maximum output voltage was obtained at a thickness of 20 cm for vertical vibration. For the transverse vibration, the maximum output voltage was obtained at a thickness of 30 cm.

(iii) The cone resistance at various penetration depths indicates that voltage output increased proportionally with the cone resistance after excitation at the natural frequency. For excitation frequencies ($f$ = 40–50 Hz), the voltage output increased from 22–45% for cone resistance in the 9–10% range. This variation indicates that the densification of fill and consequent increase in the cone resistance significantly impact improving the efficiency of the *PZT* patch.

(iv) The natural frequency and damping loss factor significantly influence the voltage output of the *PZT* patch.

**Author Contributions:** Conceptualization, N.K. and A.T.; methodology, N.K. and A.T.; formal analysis, N.K.; investigation, N.K.; resources, N.K. and A.T.; data curation, N.K.; writing—original draft preparation, N.K.; writing—review and editing N.K.; visualization, N.K. and A.T.; supervision, A.T. All authors have read and agreed to the published version of the manuscript.

**Funding:** This research received no external funding.

**Data Availability Statement:** Data is contained within the article.

**Acknowledgments:** The first author is thankful to Amit Bandyopadhyay, Washington state university, Pullman, for the constructive and valuable suggestion for writing the manuscript. The authors acknowledge resources and support from Delhi technological university, Delhi.

**Conflicts of Interest:** The authors declare no potential conflict of interest with respect to the research, authorship, and/or publication of this article.

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
