# Peer review of "The Effect of Confined Granular Soil on Embedded PZT Patches Using FFT and Digital Static Cone Penetrometer (DSCP)"

_applsci, doi:10.3390/app12199711_

Round 1
Reviewer 1 Report
In the manuscript, the effect of confined granular fill on the efficiency of embedded PZT is proposed by capturing the dynamic response of confined granular fill and voltage output. The idea of this paper is attractive. However, there are some questions that authors need to be addressed and solved:
1. Too many figures will make paper look prolix. The Figure 5 - Figure 9 can be concluded in one figure. And Figure 10 - Figure 11 can be assembled into one figure, and Figure 12-13 can be concluded in one figure.
2. In the paper, authors used lots of equations to illustrate how to analyze data, however, authors didn’t mention any calculated results. The demonstrated and tested are not well-combined, which will cause confusion.
3. Fig 5-9, The authors illustrate the frequency at the maximum amplitude is considered the natural frequency of the system. As the thickness (H) of the granular fill increased, the peak displacement declined when the frequency was higher than the natural frequency. However, the spectrum when H in 20cm is higher than H in 35cm. This caused confusion to audience. The authors should need more description when illustrating this figure.
4. Check the standard template of reference citation.
5. Figure 3. illustrated Layout of the PZT patch embedded into the confined granular fill for the vertical and transverse vibrations. However, to be more obvious, can authors also add an optical figure to show the real structure the embedded patch device.
6. The final efficiency of energy conversion of the PZT patch was not calculated and demonstrated in the paper.
Author Response
Please find the attachment for the point-by-point response to reviewer 1.

Reviewer 2 Report
This paper presented the evaluation of some parameters to the efficiency of the system. There are some issues that need to be addressed:
1. The introduction is not clear to me. I do not see a strong relation between other's work and this work, and it is lack of background for this work.
2. For the PZT patch, please provide the center frequency. Because there remains a possible reason that the detected natural frequency is not exactly the whole system, but the center or resonance frequency of the PZT patch.
3. Please clarify how the frequency changes affect the output of the external vibrator. For example, the gain for many amplifiers of the external vibrators are frequency dependent. It could lead to the error estimation.
4. Please clarify if a homogeneous vibration was generated along with both vertical and transverse directions.
5. Line 353-355, please discuss if this position is related to the natural focus of the PZT patch.
6. Figure 10 needs to improve resolution, the number of the titles need to be organized.
Author Response
Please find the attachment for the pint by point response of reviewer 2.

Round 2
Reviewer 1 Report
Authors revised the manuscript carefully. However, please increase the resolutions of the figure. Others look good.
Author Response
The authors are thankful for the reviewer’s suggestion. Please find the attached document of updated Figures.
